# Clinical Comorbidities and Videourodynamic Characteristics of Dysfunctional Voiding in Women

**DOI:** 10.3390/biomedicines13010199

**Published:** 2025-01-15

**Authors:** Cheng-Ling Lee, Yuan-Hong Jiang, Jia-Fong Jhang, Tien-Lin Chang, Jing-Hui Tian, Hann-Chorng Kuo

**Affiliations:** Department of Urology, Hualien Tzu Chi Hospital, Buddhist Tzu Chi Medical Foundation, Tzu Chi University, Hualien 97002, Taiwan; leecl@hotmail.com (C.-L.L.); redeemerhd@gmail.com (Y.-H.J.); alur1984@hotmail.com (J.-F.J.); zxc13912@gmail.com (T.-L.C.); luckysweet999@gmail.com (J.-H.T.)

**Keywords:** voiding dysfunction, urgency incontinence, pelvic floor dysfunction, urethral sphincter

## Abstract

**Purpose:** The current study aimed to investigate the clinical comorbidities and urodynamic characteristics of a large cohort of women with dysfunctional voiding (DV) validated on a videourodynamic study (VUDS). **Methods:** Women who presented with VUDS-confirmed DV from 1998 to 2022 were retrospectively analyzed. Data on clinical symptoms, VUDS findings, and medical comorbidities including medical illness and previous surgical history were recorded and examined. Patients with DV were subgrouped according to age, presence of medical comorbidity, and different urodynamic parameters. The urodynamic parameters and treatment outcomes among the different subgroups were examined. **Results:** In total, 216 women were retrospectively analyzed. Among them, 188 (88.3%) presented with storage symptoms and 130 (61.0%) with voiding symptoms. Regarding outcomes, 48 (22.2%) patients had successful treatment outcomes; 76 (35.2%), improved outcomes; and 92 (42.6%), failed outcomes. Then, 150 (69.0%) patients presented with urodynamic DO. Patients with terminal DO experienced a significantly higher incidence of hypertension (56.8%), diabetes mellitus (37.9%), and latent central nervous system diseases (38.9%) than those with non-DO or phasic DO. Patients with phasic DO had a significantly higher detrusor pressure (Pdet) and bladder outlet obstruction index than those with non-DO and terminal DO. Patients with hypertension or those with a Pdet ≥ 35 cmH_2_O had high rates of successful treatment outcomes. **Conclusions:** DV is significantly associated with older age and a higher incidence of central nervous system diseases, hypertension, and diabetes mellitus in women. Patients with phasic DO had a high Pdet and BOO, and patients with hypertension or those with Pdet ≥35 cm H_2_O who received urethral sphincter treatment had a better treatment outcome.

## 1. Introduction

Voiding dysfunction in women is not rare. The pathophysiology of voiding dysfunction can be associated with functional or anatomical etiologies. Functional disorders may be caused by detrusor underactivity, bladder hypersensitivity, bladder neck dysfunction, pelvic floor dysfunction, or urethral sphincter dysfunction, which result in a small voided volume, difficult initiation, slow stream, and incomplete bladder emptying [1]. Dysfunctional voiding (DV) is defined as non-neurogenic voiding dysfunction caused by a non-relaxing urethral sphincter or dis-coordinated urethral sphincter relaxation during voiding [2].

The actual pathophysiology of DV in women has not been completely elucidated. However, women with clinically diagnosed DV commonly exhibit bacterial cystitis, pelvic surgery, chronic constipation, or medical comorbidities [3]. Further, patients with DV may present with different lower urinary tract symptoms (LUTSs), including storage and voiding LUTSs, and characteristic videourodynamic study (VUDS) findings [4]. The association between medical comorbidity and DV has not been confirmed. However, learned habits, latent neurogenic lesions, or responses to detrusor overactivity (DO) are attributed to DV [5].

Since the pathophysiology of DV has not been fully identified, there are still no effective treatment strategies for it. The management of DV mainly aims to reduce bladder outlet resistance, including urethral sphincter relaxation or pelvic floor muscle hypertonicity [6]. Various methods such as the use of alpha-blocker and skeletal muscle relaxant, pelvic floor muscle biofeedback, urethral sphincter botulinum toxin injection, and sacral nerve stimulation have been used [7,8,9]. However, none of them had a durable and effective treatment outcome. Therefore, there could be some unknown pathophysiological mechanisms underlying DV that are not adequately treated by the currently available therapeutic modalities. If we can find an association of DV with other comorbidities or urodynamic characteristics, we might find effective treatment approaches to improve the successful rate of treatment.

The current study aimed to investigate the clinical comorbidities and urodynamic characteristics of a large cohort of women with VUDS-confirmed DV. The results could provide an overview of DV in women and explore the possible pathophysiological mechanisms underlying lower urinary tract dysfunction and further treatment strategy.

## 2. Materials and Methods

### 2.1. Study Design

This is a retrospective analysis of the clinical comorbidities and urodynamic characteristics of a large cohort of women with dysfunctional voiding validated on a videourodynamic study in a single tertiary hospital.

### 2.2. Data Source and Study Population

Women who presented with VUDS-confirmed DV from 1998 to 2021 were retrospectively analyzed. Data on the clinical symptoms, VUDS findings, and medical comorbidities including medical illness and previous surgeries were recorded and analyzed. Patients who had VUDS reports but whose medical data were missing were not included in the final analysis.

### 2.3. Socio-Demographic and Clinical Data

To investigate the association between the development of DV and medical diseases, a chart review was performed after the initial diagnosis of DV and medical diseases such as diabetes mellitus, hypertension, cardiovascular disease, chronic kidney disease, and congestive heart failure, as well as neurological diseases such as cerebrovascular disease (CVA), Parkinson’s disease (PD), and dementia. Details on previous gynecological or pelvic organ surgery were recorded. Patients with overt neurogenic lower urinary tract dysfunction, active urinary tract infection, or anatomical bladder outlet obstruction (BOO) during DV diagnosis were excluded from this study. Patients with medical records of central nervous system (CNS) diseases after the diagnosis of DV were considered to have latent CNS disease in this study.

### 2.4. Videourodynamic Study

VUDS was performed using a standard procedure with recording of intravesical pressure, intra-abdominal pressure, electromyography of the pelvic floor muscles, and concomitant cinefluoroscopic examination of the bladder and bladder outlet. The VUDS parameters included the first sensation of filling, fullness sensation (FS), urgency sensation (US), and cystometric bladder capacity (CBC) during the storage phase. Patients were allowed to void if they had a strong urge to void. The voiding detrusor pressure (Pdet), maximum flow rate (Qmax), voided volume, and post-void residual volume (PVR) were recorded during the voiding phase [10]. During voiding, cinefluoroscopic cystourethrography was performed, and the urethral patency or narrowing sites were evaluated. Patients with a Pdet < 30 cm H_2_O with a patent bladder neck and urethra during voiding were diagnosed with non-obstructive VUDS. Bladder neck obstruction was considered if the bladder neck did not have a funnel shape with or without an elevated Pdet during voiding. If patients had a Pdet ≥ 35 cm H_2_O and there was a narrow site at the midurethra during voiding, DV was considered [4]. If the Pdet was ≥35 cm H_2_O and urethral narrowing was located at the distal site, poor relaxation of the pelvic floor muscles was considered.

DV was diagnosed based on the characteristic findings on a VUDS [4]. The definitive diagnosis of DV was made if typical external sphincter contraction was observed during micturition in the course of electromyography of the pelvic floor muscles study [11], or fluoroscopic visualization of the spinning top urethra sign during voiding [12]. A VUDS is advantageous when visualizing bladder outlet narrowing during voiding with a high Pdet. During voiding cystourethrography, the VUDS characteristics of DV include a widely opened bladder neck and a dilated proximal urethra to the level of the external sphincter, which can be clearly differentiated from a primary bladder neck dysfunction or urethral stricture in a female BOO [4].

The terminology and urodynamic parameters were reported in accordance with the recommendations of the International Continence Society [10]. Phasic DO was diagnosed if phasic detrusor contraction occurred during the bladder storage phase and the patient perceived an US. If the uninhibited detrusor contraction occurred at the end of the bladder filling and the patient felt a strong urge to void, a diagnosis of terminal DO was made. According to the VUDS findings, patients were subgrouped into patients without DO (non-DO), with phasic DO, and with terminal DO. The medical comorbidities and VUDS characteristics were compared between patients with and without DO.

### 2.5. Treatment and Outcome Assessment

Patients received medical treatment or an urethral botulinum toxin A (BoNT-A) injection according to their VUDS findings. The treatment outcomes were analyzed according to different VUDS subtypes. If patients experienced an improvement in voiding symptoms and the follow-up urodynamic study revealed decreased Pdet and PVR and increased Qmax, the treatment outcome was considered successful. If the Pdet was reduced but the Qmax did not improve or the PVR remained high, the treatment outcome was considered as improved. Patients with a successful and improved treatment outcome were defined as having a satisfactory result. Treatment failure was considered if patients did not present with a reduction in Pdet or PVR, or did not experience an improvement in the Qmax.

### 2.6. Statistical Analysis

Continuous variables were expressed as means with standard deviations, and categorical data were presented as numbers and percentages (%). The chi-square test for categorical variables and the Wilcoxon rank-sum test for continuous variables were used to determine *p*-values between groups for statistical comparisons. A sensitivity analysis was performed for the comorbidity and urodynamic variables associated with the treatment outcome by excluding the patients with outliers of urodynamic variables. All assessments were two-sided, and a *p* value of <0.05 was considered statistically significant. The Statistical Package for the Social Sciences software for Windows version 16.0 (IBM Inc., Chicago, IL, USA) was used in all calculations.

### 2.7. Ethical Statement

The study was conducted in accordance with the Declaration of Helsinki, and approved by the Institutional Review Board of Hualien Tzu Chi Hospital (approval no. 110-147-B, dated 15 July 2021). The need for an informed consent was waived due to the nature of the retrospective analysis.

## 3. Results

In total, 216 women with VUDS-confirmed DV were retrospectively analyzed. The mean age of the participants was 58.5 ± 16.8 years, and most of them were aged 46–80 years. Storage symptoms, including frequency (n = 162, 76.1%), urgency (n = 152, 71.4%), urgency urinary incontinence (n = 88, 41.3%), and nocturia (n = 62, 29.1%), were reported in 188 (88.3%) women. Further, 130 (61.0%) patients presented with voiding symptoms, including urination difficulty (n = 107, 50.2%), urinary retention (n = 42, 19.7%), terminal dribble (n = 1, 0.5%), micturition pain (n = 21, 11.7%), and post-micturition residual sensation (n = 21, 9.9%). In terms of outcomes, 48 (22.2%) patients exhibited successful treatment outcomes; 76 (35.2%), improved outcomes; and 92 (42.6%), failed outcomes.

### 3.1. Medical Comorbidities in DV Patients with and Without DO

Table 1 shows the data on medical comorbidities and previous gynecological and pelvic organ surgeries in 216 patients with DV. Results showed that urodynamic DO was noted in 150 (69.4%) patients with DV, including 95 (44.0%) with terminal DO and 55 (25.5%) with phasic DO. Patients with urodynamic DO had a higher incidence of hypertension (49.0%), CVA (14.8%), and dementia (7.4%) compared with those without DO. When we divided patients with DO into subgroups of phasic DO (n = 55) and terminal DO (n = 95), patients with terminal DO were significantly older (mean age: 65.1 ± 16.5 years). Patients with terminal DO had a significantly higher incidence of hypertension (56.8%), diabetes mellitus (37.9%), and latent CNS diseases (38.9%) than those with non-DO or phasic DO (Table 2).

### 3.2. Association of Urodynamic Parameters with the Presence of DO Subtypes

Table 3 shows the urodynamic parameters of patients with DV and different DO subtypes. Patients with phasic and terminal DO experienced a significantly greater increase in bladder sensation (FSF, FS, and US) and CBC than those with non-DO. Patients with phasic DO had a significantly higher Pdet and bladder outlet obstruction index (BOOI) than those with terminal DO and non-DO. Patients with terminal DO had a significantly lower Qmax and voiding efficiency (VE) than those with phasic DO and non-DO.

### 3.3. Association of Medical Comorbidities with Urodynamic Parameters

The association between medical comorbidities and urodynamic parameters was evaluated. Results revealed that patients with hypertension, diabetes mellitus, or CNS lesion were significantly older, and also had a lower Pdet and lower Qmax, compared with the patients without these comorbidities. A significantly smaller voided volume was noted in patients with hypertension and CNS lesion. A significantly higher PVR and lower VE were noted in patients with hypertension and diabetes mellitus. A higher incidence of urodynamic DO was also noted in patients with a CNS lesion. However, there was no significant difference in the urodynamic parameters between patients who underwent gynecologic or pelvic organ surgery and those who did not (Table 4). The urodynamic results revealed that DV patients without comorbidities were younger, had a higher voiding pressure, higher Qmax, and a relatively higher BOO index.

### 3.4. Comparison of the Treatment Outcome in DV Patients with and Without Risk Factors of Voiding Dysfunction

The long-term treatment success rate was 22.2% (48/216) and the improved rate was 35.2% (76/216). We select comorbidity and urodynamic variables that are associated with DO, BOO, and voiding dysfunction, and compare the association of these risk factors with the long-term treatment outcome. Table 5 shows the treatment outcomes of patients with DV according to age, medical comorbidities, and different urodynamic parameters and DO subtypes. Patients with hypertension and those with a Pdet ≥ 35 cmH_2_O significantly differed in terms of treatment outcomes. When a successful and improved treatment outcome is considered as satisfactory, patients with hypertension (66.7%) and those with a Pdet ≥ 35 cmH_2_O (64.1%) had a higher satisfactory outcome than those without hypertension (50.8%) or those with a Pdet < 35 cmH_2_O (32.6%). The other subgroups did not significantly differ in terms of other factors such as age, diabetes mellitus, CNS disease, large PVR, low VE, or DO subtypes. We also performed a sensitivity analysis for the findings of variables on the treatment outcome. After excluding the outliers of PVR > 400 mL and Pdet > 100 cmH_2_O, a total of 197 patients were analyzed. The satisfactory rate of with or without hypertension (65.9% vs. 34.1%, *p* = 0.041) and Pdet > 35 cmH_2_O or <35 cmH_2_O (64.1% vs. 31.8%, *p* < 0.001) remains a significant variable for a satisfactory treatment outcome. The other variables do not have an effect on the treatment outcome.

## 4. Discussion

The current study showed that DV was significantly associated with urodynamic DO, hypertension, diabetes mellitus, latent CNS diseases, and previous gynecological or pelvic surgery in women. More than 70% of women with DV had urodynamic DO and urgency frequency symptoms. Women with terminal DO were significantly older than those with phasic DO or non-DO and had a higher incidence of hypertension and latent CNS diseases. Patients with hypertension and those with a higher Pdet at the baseline had a better long-term treatment outcome.

The prevalence of DV in women is not low. In particular, approximately 2–10% of women with LUTS have urodynamic-confirmed DV [4,5]. However, the pathophysiology of DV in women has not been completely elucidated. In people with a normal neurological function, the urethral sphincter should be coordinated during micturition. Micturition is triggered by the release of tonic inhibition from the suprapontine centers and the release of the trigger signal from the pontine micturition center [13]. However, patients with urodynamic DO or CNS diseases might develop a guarding reflex to inhibit urine leakage if detrusor contractions occur [14]. DV is considered to be caused by a learned behavioral disturbance in early childhood and may be treated by re-educational therapy [5]. When taking this into consideration, DO might also be a trigger factor for DV if patients used to hold back urine at each urgency episode [15]. The long-term increase in urethral sphincter activity caused by a reflex inhibitory response to the development of DO could result in spasticity of the urethral sphincter and, finally, functional BOO appearance during voiding. With a long-term learning habit, the urethral sphincter might become hyperactive or have poor relaxation during urination, resulting in clinical DV [16]. This study found that 69.4% of patients with DV had urodynamic DO. Therefore, the occurrence of DO during the bladder storage phase could play an important role in the pathogenesis of DV.

DV is frequently observed in children with urgent urinary incontinence, recurrent urinary tract infection, and vesicoureteral reflux [17]. Delayed maturation of the brain is a cause of DV in children. These patients also have frequent DO during bladder storage, resulting in a tight urethral sphincter during the voiding phase. However, in women, DV is more prevalent at 45–80 years of age, which is not likely to extend from a childhood voiding dysfunction. Therefore, the pathophysiology of DV in women might be different from that in children. Urinary tract infection and chronic constipation are associated with DV in children [18,19]. Further, this study revealed that hypertension, latent CNS diseases, diabetes mellitus, and gynecological or pelvic organ surgery contribute to the development of DV in adult women. These diseases might have an effect on bladder oversensitivity, bladder overactivity, and pelvic floor hypertonicity, and can lead to a hypertonic or hyperactive urethral sphincter. In our experience, some patients might develop upper tract function deterioration due to a low compliant contracted bladder after long-term bladder outlet obstruction. Early identification and intervention of DV can reverse the bladder overactivity and chronic inflammation due to long-term BOO, and decrease the risks of recurrent urinary tract infection and obstructive uropathy.

The results of this study revealed that female patients with DV had a high prevalence of hypertension and diabetes, and that a high number of patients had urodynamic DO. A population-based study has revealed that an overactive bladder is positively associated with metabolic syndromes, such as obesity, hyperlipidemia, hypertension, and diabetes mellitus [20]. Patients with hypertension had a significantly higher incidence of overactive bladder than those without hypertension [21]. The association of hypertension with an endothelial dysfunction, increased oxidative stress, and DO has been well documented [22]. Women with DV were found to have an increased reactive-oxidative stress response and detrusor hyperactivity [23]. These consequences link hypertension and urodynamic DO in patients with DV. The occurrence of DO during bladder storage will increase the urethral sphincter guarding response to bladder overactivity. The treatment of DV with combined alpha-adrenergic blockers and antimuscarinics has been well recommended, and might decrease the urethral sphincter resistance as well as decreasing detrusor hyperactivity to achieve a better outcome [24]. In addition, this study shows that patients with hypertension are also older. The vascular endothelial dysfunction in older women leads to a higher incidence of hypertension and is associated with a greater incidence of overactive bladder and urodynamic DO. However, the direct association of DO with DV is still hypothetical, and there is no direct evidence supporting this connection [7]. Nevertheless, the proportion of DV patients with hypertension who had a successful or improved treatment outcome was higher than that of patients without hypertension. This finding suggests an association between hypertension and DO, and a better treatment outcome of DV. This study also showed that 27.8% of patients with DV had diabetes mellitus, which was associated with a larger PVR and lower VE at the baseline. Diabetes mellitus was not related to a failed treatment outcome in this study. However, a decreased detrusor contractility and VE in patients with diabetes mellitus might result in less improvement in the voiding efficiency after the active treatment of DV [25].

Patients with latent CNS disease may develop CVA, PD, or dementia several years after the onset of voiding dysfunction [26]. Our previous study showed that >10% of elderly patients with urodynamic DV will have CNS diseases after a long-term follow-up [27]. The dis-coordinated urethral sphincter detected on VUDS in patients with voiding dysfunction might be associated with an abnormal link in the micturition reflex network in the brain and an abnormal endothelial function or possible CVA [28]. In this study, 38.9% of patients with DV and terminal DO had latent CNS diseases, and this patient subgroup was significantly older than the other subgroups. However, only 5.5% of patients presented with phasic DO and 6.1% with non-DO had CNS lesions. This result indicates that terminal DO in patients with DV might imply the presence of latent CNS diseases and the need for neurological consultation for early intervention. Interestingly, DV patients with phasic DO were found to have a significantly higher Pdet than those with non-DO or terminal DO. This result further shows that patients with phasic DO and terminal DO might have a distinct etiology of DV and result in different treatment outcomes [29]. In patients with phasic DO, DV might be caused by true urethral sphincter dysfunction, and these patients had a higher Pdet during voiding. In contrast, DV patients with a low Pdet might have been attributed to a guarding reflex caused by terminal DO and pelvic floor muscle hypertonicity, resulting in a higher incidence of failed treatment outcome.

Recent research on urinary biomarkers in women with DV revealed that patients with DV had evidently higher urine oxidative stress biomarker and inflammatory marker profiles than the controls [24]. They had elevated urine 8-OHdG and IL-1β levels, which were also positively correlated with the clinical symptoms. Based on this result, DV might be associated with increased bladder inflammation caused by BOO and bladder oxidative stress and the high prevalence of urgency frequency symptoms in women with voiding dysfunction [7]. Therefore, these urine analytes might have diagnostic and prognostic values and could be used as biomarkers of DV among women. Future research on the treatment of female DV might focus on the urine biomarker expression and divide DV into different subtypes for different treatment modalities based on the expressions of inflammatory and oxidative stress biomarkers. The treatment success might be increased.

The treatment of female DV is not always satisfactory. There is no optimal treatment strategy for it. Medical treatment with alpha-blocker, biofeedback pelvic floor training, urethral BoNT-A injections, sacral nerve stimulations, and posterior tibial nerve stimulation has been recommended [25,30,31,32,33]. Although an urethral sphincter BoNT-A injection could reduce urethral resistance, the clinical application on DV is still limited, mainly because the therapeutic efficacy has not been stably established [34]. This result further implies that the pathophysiology of DV in women is not solely attributed to the urethral sphincter muscles; CNS dysfunction might also contribute to the inadequate relaxation of the urethral sphincter during urination.

DV with different VUDS characteristics can have a different underlying pathophysiology, which may also result in different treatment outcomes. In the long-term study of women with DV, medical treatment with or without urethral BoNT-A injection resulted in a significant reduction in Pdet and BOOI. A more prominent obstructive profile at the baseline such as a higher PVR and maximal urethral closure pressure in the urodynamic study had been found to be associated with a higher successful voiding restoration [8]. The results of this study also revealed that DV patients with a higher Pdet experienced a significantly higher satisfactory rate after treatment, suggesting that the pathophysiology of DV that has a good response to treatment targeting at the urethral sphincter is different from DV refractory to treatment. Therefore, a successful treatment outcome of BoNT-A injections on the urethral sphincter is applicable for true external sphincter spasticity but not for other causes of voiding dysfunction in women. A higher treatment success for DV may be achieved only when the etiology of DV is urethral sphincter dysfunction but not the guarding effect to the development of DO. The heterogeneity of the underlying pathophysiology of DV results in limited success rates after urological treatment.

The strength of this study is a large cohort of female patients with DV, and the diagnosis of DV was validated by the comprehensive videourodynamic study. Because the study was performed in a single medical center, most patients’ medical records, treatment modalities, and treatment outcomes can be assessed after a long-term follow-up. Moreover, the treatment outcomes were analyzed according to different DO subtypes and urodynamic parameters. The data provide valuable insights into the pathophysiology of DV and treatment strategy. However, some limitations that might affect the results of the study need to be addressed. First, selection bias and recall bias are possible because some patients who were lost to follow-up or did not have medical records in this hospital could be missed. Second, this study did not use validated questionnaires to assess the symptoms, which might affect the reliability of patients’ reported symptoms. Third, the study population may not be representative of all women with voiding dysfunction because the incidence of DV is around 10% of the women with a voiding dysfunction [7,29]. When a woman has a voiding dysfunction refractory to treatment, the VUDS should be investigated to search for the underlying pathophysiology. Medical comorbidity should be concomitantly treated in addition to the management targeting at the urethral sphincter hypertonicity.

## 5. Conclusions

The pathophysiology of female DV has not been fully elucidated. This study revealed that DV is significantly associated with older age and a higher incidence of CNS diseases, hypertension, and diabetes mellitus in women, particularly those with terminal DO. Patients with phasic DO had a higher Pdet and BOOI but a better outcome with urethral sphincter treatment; therefore, the pathophysiology of DV might be urethral sphincter dysfunction. Patients with a Pdet < 35 cmH_2_O had a higher treatment failure rate. Therefore, urethral sphincter discoordination might be a result of DO, not primarily of urethral sphincter dysfunction.

## Figures and Tables

**Table 1 biomedicines-13-00199-t001:** Baseline demographic and clinical characteristics of patients with dysfunctional voiding according to detrusor overactivity.

	Total (n = 216)	Non-DO (n = 66)	DO (n = 150)	*p* Value
Age	58.7 ± 16.8	53.7 ± 15.7	60.9 ± 16.8	0.004
Medical disease				
Hypertension	90 (41.7%)	17 (25.8%)	73 (48.7%)	0.009
Diabetes mellitus	63 (29.2%)	14 (21.2%)	49 (32.7%)	0.168
Coronary arterial disease	21 (9.7%)	6 (9.1%)	15 (10.0%)	0.877
Congestive heart failure	15 (6.9%)	2 (3.0%)	13 (8.7%)	0.241
Chronic kidney disease	17 (7.9%)	5 (7.6%)	12 (8.0%)	0.953
Parkinson’s disease	7 (3.2%)	1 (1.5%)	6 (4.0%)	0.677
Cerebrovascular disease	25 (11.6%)	3 (4.5%)	22 (14.7%)	0.036
Dementia	11 (5.1%)	0 (0.0%)	11 (7.3%)	0.036
Cerebral palsy	1 (0.5%)	0 (0.0%)	1 (0.7%)	1.000
Immune disease	10(4.7%)	2 (3.1%)	8 (5.4%)	0.727
Uterine myoma	10 (4.7%)	5 (7.8%)	5 (3.4%)	0.171
Ovarian cyst	16 (7.5%)	6 (9.4%)	10 (6.7%)	0.572
Adenomyosis	6 (2.8%)	2 (3.1%)	4 (2.7%)	1.000
Cervical cancer	12 (5.6%)	4 (6.3%)	8 (5.4%)	0.755
Colon cancer	6 (2.8%)	3 (4.7%)	3 (2.0%)	0.368
Previous surgery				
Hysterectomy	35 (16.4%)	15 (23.4%)	20 (13.4%)	0.071
Colectomy	7 (3.3%)	2 (3.1%)	5 (3.4%)	1.000
Spinal surgery	12 (5.6%)	6 (9.4%)	6 (4.0%)	0.191

DO: detrusor overactivity.

**Table 2 biomedicines-13-00199-t002:** Baseline demographic and clinical characteristics of patients with dysfunctional voiding according to different subtypes of detrusor overactivity.

	Total (N = 216)	Non-DO (N = 66)	Phasic DO (N = 55)	Terminal DO (N = 95)	*p* Value
Age	58.7 ± 16.8	53.7 ± 15.7	53.6 ± 15.0	65.1 ± 16.5	0.000
Hypertension	90 (41.7%)	17 (25.8%)	19 (34.5%)	54 (56.8%)	0.001
Diabetes mellitus	63 (29.2%)	14 (21.2%)	13 (23.6%)	36 (37.9%)	0.066
CNS disease	44 (20.4%)	4 (6.1%)	3 (5.5%)	37 (38.9%)	0.000
Gynecological or pelvic surgery	50 (23.1%)	19 (28.8%)	13 (23.6%)	18 (18.9%)	0.308

DO: detrusor overactivity, CNS: central nervous system.

**Table 3 biomedicines-13-00199-t003:** Urodynamic parameters of patients with dysfunctional voiding according to different subtypes of detrusor overactivity.

	Total (N = 216)	(A) Non-DO (N = 66)	(B) Phasic DO (N = 55)	(C) Terminal DO (N = 95)	*p* Value	Post Hoc
Age (years)	58.7 ± 16.8	53.7 ± 15.7	53.6 ± 15.0	65.1 ± 16.5	0.000	A v C; B v C
FSF (mL)	117 ± 64.8	145 ± 85.4	98.5 ± 41.6	109 ± 53.2	0.000	A v B; A v C
FS (mL)	177 ± 85.5	220 ± 96.8	164 ± 69.0	156 ± 75.7	0.000	A v B; A v C
US (mL)	211 ± 102	269 ± 106	193 ± 85.5	182 ± 92.2	0.000	A v B; A v C
Pdet (cmH_2_O)	51.2 ± 26.7	46.8 ± 17.0	60.7 ± 30.5	49.0 ± 28.5	0.007	A v B; B v C
Compliance/FS	54.2 ± 60.9	72.2 ± 76.9	33.5 ± 25.5	54.0 ± 60.0	0.002	A v B
Compliance/Cap	55.6 ± 68.4	81.8 ± 97.6	33.4 ± 24.0	50.7 ± 55.6	0.000	A v B; A v C
Qmax (mL/s)	8.76 ± 5.27	9.14 ± 5.44	10.1 ± 4.94	7.71 ± 5.18	0.022	B v C
Volume (mL)	177 ± 114	228 ± 123	185 ± 101	137 ± 102	0.000	A v C; B v C
PVR (mL)	101 ± 117	109 ± 115	84.6 ± 122	104 ± 116	0.486	
CBC (mL)	278 ± 141	337 ± 132	269 ± 158	242 ± 123	0.000	A v B; A v C
VE	0.68 ± 0.30	0.71 ± 0.28	0.76 ± 0.27	0.62 ± 0.32	0.017	B v C
BOOI	33.7 ± 29.7	28.1 ± 21.0	40.5 ± 32.3	33.6 ± 32.5	0.074	

DO: detrusor overactivity, FSF: first sensation of filling, FS: fullness sensation, US: urgency sensation, Pdet: detrusor pressure, Qmax: maximum flow rate, PVR: post-void residual volume. CBC: cystometric bladder capacity, VE: voiding efficiency, BOOI: bladder outlet obstruction index.

**Table 4 biomedicines-13-00199-t004:** Urodynamic parameters of dysfunctional voiding according to medical comorbidity.

UDS Parameter		Latent CNS Lesion (N = 44)	Hypertension (N = 90)	Diabetes Mellitus (N = 63)	Gynecology Pelvic op (N = 50)
Age (years)	Yes No	71.8 ± 11.7 * 55.7 ± 16.4	69.9 ± 10.2 * 50.2 ± 15.8	67.9 ± 10.8 * 54.8 ± 17.4	55.5 ± 15.7 59.7 ± 17.0
FSF (mL)	Yes No	123.0 ± 56.3 115.3 ± 66.1	123.1 ± 60.0 112.4 ± 68.1	128.6 ± 59.6 112.0 ± 66.4	117.5 ± 61.0 116.9 ± 66.1
FS (mL)	Yes No	172.5 ± 67.2 178.1 ± 88.4	176.2 ± 74.9 178.4 ± 93.1	179. 8 ± 75.0 176.5 ± 89.9	182.1 ± 87.9 176.0 ± 85.0
US (mL)	Yes No	187.9 ± 76.6 215 ± 105.2	203.7 ± 87.7 215.9 ± 111.6	206.1 ± 91.4 212.6 ± 106.3	218 ± 101.7 208 ± 102.2
Pdet (cmH_2_O)	Yes No	44.0 ± 15.1 * 52.5 ± 28.2	47.0 ± 18.1 * 54.5 ± 31.4	45.7 ± 15.96 * 53.6 ± 29.9	49.5 ± 21.3 51.8 ± 28.2
Compliance. FS	Yes No	64.3 ± 67.7 52.8 ± 59.9	58.7 ± 6.46 50.8 ± 58.1	54.3 ± 56.5 54.2 ± 62.9	47.5 ± 50.9 56.3 ± 63.7
Compliance. Cap	Yes No	61.6 ± 77.6 55.0 ± 67.2	57.5 ± 68.9 54.1 ± 68.3	52.3 ± 58.8 57.0 ± 72.3	52.4 ± 71.5 56.5 ± 67.6
Qmax (mL/s)	Yes No	6.95 ± 4.75 * 9.11 ± 5.29	7.89 ± 4.61 * 9.42 ± 5.66	8.00 ± 4.80 * 9.08 ± 5.45	9.30 ± 4.30 8.59 ± 5.54
Volume (mL)	Yes No	128.1 ± 96.2 * 187.9 ± 115	159.1 ± 109.7 * 190.5 ± 116.6	161.3 ± 108.8 183.7 ± 116.5	182.6 ± 115 175 ± 114.5
PVR (mL)	Yes No	106 ± 111.7 98.6 ± 117.9	122.9 ± 121.0 * 83.6 ± 111.4	142.0 ± 137.8 * 82.8 ± 102.3	95.0 ± 112.6 102 ± 118.6
CBC (mL)	Yes No	234 ± 102.9 * 286.5 ± 145	282 ± 125.5 274.1 ± 151.9	303.3 ± 153.9 266.4 ± 133.9	277.6 ± 135 277.5 ± 143
VE	Yes No	0.60 ± 035 0.70 ± 0.29	0.59 ± 0.31 * 0.75 ± 0.27	0.57 ± 0.32 * 0.73 ± 0.28	0.69 ± 0.30 0.68 ± 0.30
BOOI	Yes No	30.1 ± 19.4 34.3 ± 31.2	31.2 ± 21.3 35.6 ± 34.7	29.7 ± 17.9 35.4 ± 33.38	30.9 ± 22.4 34.6 ± 31.6
Urodynamic DO	Yes No	33 (89.2%) * 117 (66.5%)	73 (79.3%) 76 (51.0%)	49 (76.6%) 100 (67.1%)	31 (60.8%) 118 (71.5%)

CNS: central nervous system, DO: detrusor overactivity, FSF: first sensation of filling, FS: fullness sensation, US: urgency sensation, Pdet: detrusor pressure, Qmax: maximum flow rate, PVR: post-void residual volume. CBC: cystometric bladder capacity, VE: voiding efficiency, BOOI: bladder outlet obstruction index. Asterisks indicate significant difference between patient group with and without the urodynamic variable.

**Table 5 biomedicines-13-00199-t005:** Treatment outcome of dysfunctional voiding according to medical comorbidities, different urodynamic parameters, and subtypes of detrusor overactivity.

Variables	Patient (n = 216)	Successful (n = 48)	Improved (n = 76)	Failed (n = 92)	*p* Value
Age ≥ 65 years Age < 65 years	94 122	19 (20.2%) 29 (23.8%)	36 (38.3%) 40 (32.8%)	39 (41.5%) 53 (43.4%)	0.668
Hypertension No hypertension	90 126	26(28.9%) 22(17.5%)	34 (37.8%) 42 (33.3%)	30 (33.3%) 62 (49.2%)	0.039
Diabetes mellitus No diabetes	63 153	9 (14.3%) 39 (25.5%)	29 (46.0%) 47 (30.7%)	25 (39.7%) 67 (43.8%)	0.059
CNS lesion No CNS lesion	44 172	10 (22.7%) 38 (22.1%)	16 (36.4%) 60 (34.9%)	18 (40.9%) 74 (43.0%)	0.968
Pdet ≥ 35 cmH_2_O Pdet < 35 cmH_2_O	170 46	45 (26.5%) 3 (6.5%)	64 (37.6%) 12 (26.1%)	61 (35.9%) 31 (67.4%)	<0.001
PVR ≥ 100 mL PVR < 100 ml	97 119	21 (21.6%) 27 (22.7%)	34 (35.1%) 42 (35.3%)	42 (43.3%) 50 (42.0%)	0.977
VE ≥ 0.67 VE < 0.67	128 88	29 (22.7%) 19 (21.6%)	47 (36.7%) 29 (33.0%)	52 (40.6%) 40 (45.5%)	0.770
Non DO Phasic DO Terminal DO	66 55 95	14 (21.2%) 15 (27.3%) 19 (20.0%)	22 (33.3%) 17 (30.9%) 37 (38.9%)	30 (45.5%) 23 (41.8%) 39 (41.1%)	0.774

CNS: central nervous system, Pdet: detrusor pressure, PVR: post-void residual, VE: voiding efficiency, DO: detrusor overactivity.

## Data Availability

Data is contained within the article.

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
