# Peer review of "Clinical Comorbidities and Videourodynamic Characteristics of Dysfunctional Voiding in Women"

_biomedicines, 2025, doi:10.3390/biomedicines13010199_

Round 1
Reviewer 1 Report
Comments and Suggestions for Authors
This study appears to be a well-designed retrospective analysis of a large cohort of women with dysfunctional voiding (DV). The use of videourodynamic study (VUDS) for validation adds strength to the study.
However, there are a few limitations to consider:
While retrospective studies can be useful for identifying trends and associations, they are susceptible to potential biases and confounding factors.As with any retrospective study, there is a risk of selection bias and recall bias. The authors should discuss these limitations and how they might affect the results.
How did you diagnose the underlying diseases for exclusion?
The study did not involve randomization, which limits the ability to draw causal conclusions about the relationship between different variables.
The reliance on patient-reported symptoms can introduce some degree of variability and bias.
What are the long-term consequences of untreated DV in women?
Can early identification and intervention improve outcomes for women with DV?
Are there specific biomarkers or genetic factors that can predict the risk of DV?
What are the optimal treatment strategies for different subtypes of DV?
While a control group may not be feasible in a retrospective study, a comparison with a normative population could provide additional context.
Patient-reported symptoms, such as urgency and frequency, can be subjective. The use of validated symptom questionnaires could improve the reliability of symptom assessment.
The study population may not be representative of all women with voiding dysfunction. The authors should discuss the generalizability of their findings.
While the need for informed consent was waived due to the retrospective nature of the study, it's important to ensure that patient privacy and confidentiality are protected.
The authors should discuss the quality of the data and the strategies used to handle missing data.
Performing a sensitivity analysis can help to assess the robustness of the findings and identify potential biases.
The discussion could be strengthened by more explicitly stating the specific hypotheses that were tested in the study.
While the study found associations between DV and various factors, it is important to be cautious about drawing causal conclusions. The discussion could be more nuanced in this regard.
The authors could discuss potential future research directions to further elucidate the pathophysiology of DV and to develop more effective treatments.
Author Response
Reviewer #1
This study appears to be a well-designed retrospective analysis of a large cohort of women with dysfunctional voiding (DV). The use of videourodynamic study (VUDS) for validation adds strength to the study. However, there are a few limitations to consider:
While retrospective studies can be useful for identifying trends and associations, they are susceptible to potential biases and confounding factors. As with any retrospective study, there is a risk of selection bias and recall bias. The authors should discuss these limitations and how they might affect the results.
Reply: Thank you for the comment. The patients in this study were collected from those who had VUDS confirmed dysfunctional voiding (DV) and received treatment in our hospital. (Lines 70-71) Selection bias and recall bias are possible because some patients who were lost to follow-up or did not have medical records in this hospital could be missed. These biases might affect the results. We have added this statement in the limitation of the study. (Lines 362-364)
How did you diagnose the underlying diseases for exclusion?
Reply: Thank you for the comment. We used VUDS for determining the bladder and bladder outlet dysfunctions of female voiding dysfunction. The VUDS characteristics of DV in women is distinct. (Lines 110-113) Patients with neurogenic voiding dysfunction, bladder neck obstruction, and anatomical bladder outlet obstruction were excluded. (Lines 82-84)
The study did not involve randomization, which limits the ability to draw causal conclusions about the relationship between different variables.
Reply: Thank you for the comment. Because this is a retrospective analysis of clinical and urodynamic characteristics of female DV, randomization of patient subgroups was not necessary.
The reliance on patient-reported symptoms can introduce some degree of variability and bias.
Reply: Thank you for the comment. The treatment outcomes were assessed by both symptomatic improvement but also the results of pressure flow study. (Lines 126-133)
What are the long-term consequences of untreated DV in women?
Reply: Thank you for the comment. Patients with untreated DV usually had persistent storage and voiding symptoms. In our experience, some patients might develop upper tract function deterioration due to a low compliant contracted bladder after long-term bladder outlet obstruction. (Lines 271-273)
Can early identification and intervention improve outcomes for women with DV?
Reply: Thank you for the comment. Early identification and intervention of DV can reverse the bladder overactivity and chronic inflammation due to long-term BOO, decrease the risks of recurrent urinary tract infection and obstructive uropathy. (Lines 273-275)
Are there specific biomarkers or genetic factors that can predict the risk of DV?
Reply: Thank you for the comment. Currently, the significant urine biomarkers are related with chronic inflammation and oxidative stress biomarkers due to chronic bladder outlet obstruction. (Lines 323-325) Genetic factors of DV have not been investigated.
What are the optimal treatment strategies for different subtypes of DV?
Reply: Thank you for the comment. There is no optimal treatment strategy for DV. Medical treatment with alpha-blocker, biofeedback pelvic floor training, urethral BoNT-A injections, sacral nerve stimulations, and posterior tibial nerve stimulation have been recommended. (Lines 335-338)
While a control group may not be feasible in a retrospective study, a comparison with a normative population could provide additional context.
Reply: Thank you for the comment. Because the patients were collected retrospectively, we cannot find another patient group for comparison of the medical comorbidities and treatment outcome. Therefore, we compared the medical comorbidities, VUDS characteristics, and treatment outcome between the DV patients with different subtypes of detrusor overactivity, and with and without co-morbidities.
Patient-reported symptoms, such as urgency and frequency, can be subjective. The use of validated symptom questionnaires could improve the reliability of symptom assessment.
Reply: Thank you for the comment. We agree that validated questionnaire such as IPSS or OABSS can improve the reliability of symptom assessment. However, because the study is retrospective, not all patients had records of these validated questionnaires. We have added this point in the limitation of the study. (Lines 369-372)
The study population may not be representative of all women with voiding dysfunction. The authors should discuss the generalizability of their findings.
Reply: Thank you for the comment. We have add a statement in the last paragraph of the discussion to generalize the findings to women with voiding dysfunction. (Lines 372-377)
While the need for informed consent was waived due to the retrospective nature of the study, it's important to ensure that patient privacy and confidentiality are protected.
Reply: Thank you for the comment. The study was conducted in accordance with the Declaration of Helsinki, and approved by the Institutional Review Board of Hualien Tzu Chi Hospital. (Lines 147-149)
The authors should discuss the quality of the data and the strategies used to handle missing data.
Reply: Thank you for the comment. Patients who had VUDS reports but the medical data missed were not included in the final analysis. (Lines 72-74)
Performing a sensitivity analysis can help to assess the robustness of the findings and identify potential biases
Reply: Thank you for the comment. This study is a retrospective analysis of a cohort of patients for the association of female DV with clinical comorbidities and videourodynamic characteristics. Because the pathophysiology of DV has not been fully elucidated, we try to explore the underlying pathophysiology of DV and the causes of the limited successful treatment outcome. We performed sensitivity analysis for the comorbidity and urodynamic variables on the treatment outcome, accordingly. (Lines 139-141; and lines 223-228)
The discussion could be strengthened by more explicitly stating the specific hypotheses that were tested in the study.
Reply: Thank you for the comment. We have added the aim of the study and hypothesis to be found in this study. If we can find association of DV with other comorbidities or urodynamic characteristics, we might find effective treatment approaches to improve the successful rate of treatment. (Lines 55-57 and lines 58-61)
While the study found associations between DV and various factors, it is important to be cautious about drawing causal conclusions. The discussion could be more nuanced in this regard.
Reply: Thank you for the comment. The study found that DV was significantly associated with urodynamic DO, hypertension, diabetes mellitus, latent CNS diseases, and previous gynecological or pelvic surgery in women. Patients with hypertension and a higher Pdet at baseline had a better treatment outcome. (Lines 234-236) We have discussed this finding in the discussion section. (Lines 277-321)
The authors could discuss potential future research directions to further elucidate the pathophysiology of DV and to develop more effective treatments.
Reply: Thank you for the comment. We have added a statement of the potential future researches of DV. Future researches on the treatment of female DV might focus on the urine biomarker expression and divide DV into different subtypes for different treatment modality based on the expressions of inflammatory and oxidative stress biomarkers. The treatment success might be increased. (Lines 330-333)
Reviewer 2 Report
Comments and Suggestions for Authors
Clinical Comorbidities and Videourodynamic Characteristics of Dysfunctional Voiding in Women
Dear authors:
Thank you for submitting your article to “Biomedicines”. The work shows an interesting contribution to the knowledge about clinical comorbidities and urodynamic characteristics of a large cohort of women with dysfunctional voiding validated on videourodynamic study. Please attend the following suggestions and corrections:
-Authors (self-citation)
A higher self-citation has been identified in the work. It is necessary to reduce the number of self-references to a maximum of 2 (two).
-Materials and Methods
Material and methods need an in-deep description and development.
The corresponding type of study should be included in a new subsection (e.g. “Study design”). For this, it is considered appropriate to break down the methodology, at least with the following sections: Study Design, Data Source and Study Population, Socio-demographic and Clinical Data, Statistical Analysis, and Ethical Statement. Other sections could be added for a better methodological description.
-Results.
The writing of the results is sometimes disjointed or too simple. An effort to consolidate coherence between paragraphs is required, since the reader could lose the meaning of what is intended to be conveyed. A broad development of results is mandatory.
-Discussion.
A deep writing is required to go into the research topic, the interpretation of the results (new and expanded), and their comparison with related studies by the same authorship (to a lesser extent) or by different authors.
-Others.
What are the notable milestones of your research?
The study must include its strengths as well as limitations.
Thank you.
Author Response
Reviewer #2
Thank you for submitting your article to “Biomedicines”. The work shows an interesting contribution to the knowledge about clinical comorbidities and urodynamic characteristics of a large cohort of women with dysfunctional voiding validated on videourodynamic study. Please attend the following suggestions and corrections:
-Authors (self-citation)
A higher self-citation has been identified in the work. It is necessary to reduce the number of self-references to a maximum of 2 (two).
Reply: Thank you for the comment. According to the editor’s suggestion, we have reduced the self-citation number of references to 7. Because we have published the most articles on female DV, self-citation for the diagnosis and treatment of DV are inevitable.
-Materials and Methods
Material and methods need an in-deep description and development.
The corresponding type of study should be included in a new subsection (e.g. “Study design”). For this, it is considered appropriate to break down the methodology, at least with the following sections: Study Design, Data Source and Study Population, Socio-demographic and Clinical Data, Statistical Analysis, and Ethical Statement. Other sections could be added for a better methodological description.
Reply: Thank you for the comment. We have divided the Methods to several different sections with suitable subheadings, accordingly.
-Results.
The writing of the results is sometimes disjointed or too simple. An effort to consolidate coherence between paragraphs is required, since the reader could lose the meaning of what is intended to be conveyed. A broad development of results is mandatory.
Reply: Thank you for the comment. We have divided the results into different section with individual subheading for the content. We also expanded the statement of results to make them clearer for readers to follow.
-Discussion.
A deep writing is required to go into the research topic, the interpretation of the results (new and expanded), and their comparison with related studies by the same authorship (to a lesser extent) or by different authors.
Reply: Thank you for the comment. We have revised the discussion section. The discussion is in sequence of: (1) Key findings of this study and clinical implication (Lines 234-2240), (2) pathogenesis of DV and association with urodynamic DO (Lines 242-259), (3) the comorbidities of hypertension and diabetes mellitus in women are associated with DO, and may associate with DV. (Lines 277-302), (4) Latent CNS lesion and association with DV (Lines 304-321), (5) Recent biomarker researches of DV and future research topics (Lines 323-333), (6) Treatment outcome and DV subtypes based on VUDS findings (Lines 335-344), (7) Strength and limitations of the study. (Lines 362=377)
-Others.
What are the notable milestones of your research? The study must include its strengths as well as limitations.
Reply: Thank you for the comment. We have added the strength and limitations of the study. (Lines 362-369)
Round 2
Reviewer 1 Report
Comments and Suggestions for Authors
-
Reviewer 2 Report
Comments and Suggestions for Authors
-Authors (self-citation):
A higher self-citation (7) has been identified in the work, this is unacceptable. It is necessary to reduce the number of self-references to a maximum of 2 (two). This matter was indicated in the previous review. Higher self-citation is a bad practice.
Author Response
Dear Editor:
Thank you for the comment.
We have reduced the self citation rate of reference to 2, accordingly.
Round 3
Reviewer 2 Report
Comments and Suggestions for Authors
Thank you.